# Compliance with the WHO-recommended infant feeding practices among HIV-positive mothers at Public Health Facilities of Mekelle, Tigray, northern Ethiopia

Haftom Tadesse Alemayohu[1], Solomon Weldemariam Gebrehiwot [2]*, Hadgay Hagos[2]

**1** Dabaguna Primary Hospital, Dabaguna, Tigray, Ethiopia, **2** Mekelle University, College of Health Sciences, Department of Midwifery, Mekelle, Tigray, Ethiopia

\* mikiass1708@gmail.com

## Abstract

Breastfeeding is crucial for the health of Human Immunodeficiency Virus (HIV)-exposed infants, survival, and development. In the absence of interventions, approximately 10–20% of children can acquire HIV infection during breastfeeding. There is a dearth of studies on this topic, and the findings are inconsistent in the study region. Therefore, the aim of this study is to assess compliance with the World Health Organization (WHO) recommended infant feeding practices among HIV-positive mothers at public health facilities in Mekelle city, Tigray, northern Ethiopia. A health facility-based cross-sectional study was conducted at Mekelle city. A total of 283 HIV-positive mothers attending the prevention of mother-to-child transmission service were included in this study. Data were collected through a face-to-face interview technique. Data entry and analysis were performed using Statistical Package for Social Sciences (SPSS) version 21. Both crude and adjusted odds ratios were computed, and the level of significance was declared based on the adjusted odds ratio with a 95% confidence interval at a p-value < 0.05. In this study, the proportion of WHO- recommended infant feeding practice was 79% with a 95% CI (74.2-82.1%). Having antenatal care (ANC) follow-up (AOR = 4.92, 95% CI: 1.2- 20.7), mode of delivery (AOR = 3.04, 95% CI: 1.5 - 6.32), disclosure of HIV status (AOR = 2.72, 95% CI: 1.23 - 6.02), and having good attitude (AOR = 4.31, 95% CI: 1.91 - 9.8) were significantly associated with the outcome variable. In conclusion, the study found that a large proportion of HIV-positive mothers implemented the recommended infant feeding practice. Antenatal care follow-up, mode of delivery, HIV disclosure status, and maternal attitude were factors associated with infant feeding practice. Therefore, various actors in the health system should develop strategies that promote infant feeding practices appropriately to reduce the risk of mother-to-child transmission of HIV infection.

**Data availability statement:** All relevant data are within the paper and its Supporting Information files.

**Funding:** This research was supported by Mekelle University postgraduate council with a grant number of CRPO/CHS/SM/008/11 to HT for data collection. The funder had no role in study design, data collection, and analysis, decision to publish, or preparation of the manuscript.

**Competing interests:** The authors have declared that no competing interests exist.

## Background

In resource-limited regions, access to proven interventions for the prevention of Mother-to-Child Transmission (PMTCT) is available. However, the HIV epidemic among children still continues as a public health burden [1]. Similar to the general population, breastfeeding is crucial for the health of HIV-exposed infants, survival, and development [2–4]. HIV infection is one of the serious public health problems worldwide [5]. Globally, among the 39.0 million people living with HIV at the end of 2022, 1.5 million of them were children (0–14 years old) [6]. Moreover, among the 630,000 registered deaths related to HIV morbidity, 84,000 of them were children [6]. The tragic burden of HIV is particularly high in Sub-Saharan Africa (SSA), and this region is recognized as a place for about 13.2 million (90%) of children infected with HIV [7, 8]. According to the 2021 Ethiopian Public Health Institute (EPHI) report, the number of children living with HIV, and annual AIDS deaths in Ethiopia and Tigray were (41,788, 4, 287) and (1,892, 181), respectively [9]. In the absence of interventions, approximately 15–30% of children acquire HIV infection from their mothers during pregnancy and intrapartum, and an additional 10–20% are acquired during breastfeeding [10]. In SSA, Mother-to-Child Transmission (MTCT) of HIV accounted for 90% of pediatric infection [7, 11-13]. However, according to the UNAIDS recommendation in 2021, the rate of MTCT should not be more than 5% and 2% in countries where breastfeeding and replacement feeding are commonly practiced, respectively, as part of the elimination program target [14]. Breastfeeding close to universal level can prevent 823,000 annual deaths among under-5 children and 20,000 annual deaths from breast cancer among women [15]. It also reduces type 2 diabetes by 35% and obesity by 13% [4].

Distinguishing the considerable advantage of breastfeeding for the health of an infant and reducing the risk of postnatal HIV transmission, the 2016WHO guideline recommended breastfeeding to be early and exclusive for the first 6 months, followed by continued breastfeeding for at least 12 months and may continue for up to 24 months of life or longer with the introduction of complementary foods at six months [16]. Even though it is recognized that HIV can be transmitted via breastfeeding, the risk of morbidity and mortality is also higher among the non-breastfeeding infants caused by diarrhea, pneumonia, and malnutrition [2]. However, the risk of transmission through breastfeeding can be reduced through effective antiretroviral therapy (ART) to 1% or less [3, 17-18]. Replacement feeding is more effective at preventing the risk of MTCT of HIV during postpartum when it is acceptable, feasible, affordable, sustainable, and safe (AFASS criteria) [16]. Globally, approximately 300, 000 children are infected with HIV through breastfeeding per year. However, non-breastfeeding resulted in 1.5 million deaths in children [19]. Non-breastfeeding is also associated with $302 billion or 0.49% of the world's gross national income and annual economic losses [20]. PMTC coverage and the vertical transmission of HIV were reported at 79.5% and 14.7% in the Tigray region [9], and the level of MTCT in Mekelle public hospitals was 3.6% [21].

Even though breastfeeding is a tradition in Ethiopia, variation still continues from study to study, ranging from 49.3% in Wolayta [22] to 88.8% in Tigray, Ethiopia [23].

Moreover, reports from other regions of Ethiopia, Oromia, and Amhara revealed that the proportion of appropriate infant feeding was 85.5% and 89.5%, respectively [24–26]. About 74% and 51% of mothers had knowledge that HIV can be transmitted via breastfeeding and the role of ART in reducing the risk of vertical transmission of HIV effectively, respectively [27]. Several variables from previous published studies, such as educational status, occupational status, age, residence [25,26,28,29], antenatal care (ANC) contact, post-natal visit, use of ART, disclosure status to partner, place of delivery, mode of delivery, infant mouth ulcer, breast problem, received counseling about infant feeding options from health care providers, knowledge and attitude towards infant feeding [24-26,28-34] were factors significantly associated with infant feeding practice among HIV positive mothers.

To ensure HIV-free survival of HIV exposed infants, the government of Ethiopia has taken a number of interventions, such as the development of the Option B+ guideline, National Road Map on HIV, and National guideline for PMTCT of HIV, Syphilis, and Hepatitis B Virus [5, 35]. In addition, the country also developed the HIV/AIDS National Strategic Plan (2021–2025) to attain a reduction of new HIV infection among children from 13.39% to less than 5% by 2025; and less than 2% by 2030 [36]. As a result of these efforts, the proportion of HIV infection among children declined from 10.9% in 2012 to 4.0% in 2016 in Ethiopia [37]. MTCT also reduced from 14.7% in 2021 to 7.7% in 2022 [9].

Despite these improvements made, gaps still continue in preventing vertical HIV infection [38]. Ethiopia remains one of the top ten countries in the world with its highest burden of HIV infection among children due to MTCT [39]. Hence, to create an HIV-free generation, countries need updated and consistent information as well as updated strategies on infant feeding practice among HIV-exposed infants. Studies conducted in the study region are few in number, and they have inconsistent findings. Moreover, updated information after the devastating war in Tigray is also imperative for policy-makers. Therefore, the aim of this study is to assess the proportion of appropriate infant feeding practice and influencing factors among women living with HIV in Mekelle health facilities.

## Methods

### Study setting, design and period

A health facilities-based cross-sectional study was conducted in Tigray Regional State in Mekelle city, from March 1 to April 30/2021. Mekelle, the capital city of the Tigray regional state is, located 783 kms north of Addis Ababa, the capital city of Ethiopia. Based on the 2007 national census conducted by the Central Statistical Agency of Ethiopia (CSA), Mekelle city had a total population of 233,012 composed of 48.2% men and 51.8% women [40]. Mekelle city has 7 sub cities, 18 "kebelles" and 74 "ketenas". The city has one comprehensive and specialized hospital, two public general hospitals, one primary hospital, one military defense general hospital, and ten health centers. The prevalence of HIV/AIDS in urban areas of Tigray was 2.7% [41]. According to the unpublished regional health bureau report, there were about 898 HIV exposed infants in Mekelle city in June, 2020. The study was conducted from March 1to April 30/ 2021.

### Population

Mothers who had HIV-exposed infants and attended PMTCT services in all public health institutions of Mekelle city were considered the source of population, and mothers who had HIV-exposed infants and enrolled in PMTCT service at the selected health facilities during the data collection period were considered the study population in this study. Women living with HIV who had HIV-exposed infants less than 18 months age, enrolled in PMTCT follow-up were taken as eligibility criteria for participation.

### Sample size determination and sampling process

Initially, sample size was calculated using a single population proportion formula ($n = Z^2 p (1- p)/d^2$) for the first objective by considering the following assumptions: 95% of confidence level, 5% margin of error, and the proportion of appropriate infant feeding practice, 85.5% which was taken from a study done in Oromia region, Ethiopia [24]. After adding a 10%

non-response rate, the sample size was 210. For the second objective, the StatCalc function of Epi-Info 7.2 software was used to calculate the sample size. Variables that made significant association with appropriate infant feeding practice were used to calculate the sample size from the same study. Assumptions used in this calculation were: confidence interval of 95%, margin of error 5%, power of the study 80%, percent outcome of the unexposed and exposed groups were 41.3% and 21.5%, respectively, odds ratio 0.39 with a ratio of unexposed to exposed 5.15. Therefore, among the seven variables (maternal attitude towards breastfeeding, infant mouth ulcer, having maternal breast problem, disclosure of HIV status to partner, receiving counseling on infant feeding options, having ANC and postnatal visits) which made association with appropriate infant feeding practice, attitude of participants was produced the highest sample size (360) as compared to the sample size produced from the initial calculation using objective one and the rest associated variables included in the calculation using the second objective. However, the source population was below 10,000, as a result sample size correction formula was applied (n= (n/1+ (n/N)), n= (360/1+ (360/898) =257 and by adding 10% non-response rate (26+257), the final sample size was 283.

## Sampling procedure

There was a total of 15 health facilities (HFs) in Mekelle City, among which 8 HFs were selected for this study using a lottery method. Mekelle city has 5 hospitals (2 general, 1 defense general hospital, 1 comprehensive specialized, and 1 primary) and 10 health centers. Five health centers were randomly selected, and two general hospitals (Mekelle and Quiha general hospital), and the specialized hospital were included in this study due to their high flow of PMTCT attendees. The two hospitals (Defense and primary hospitals) were excluded because these hospitals did not have PMTCT service at that time. The total sample size was proportionally distributed to each HF based on the number of PMTCT attendees. The total number of women living with HIV and who had HIV-exposed infants enrolled in the PMTCT service in the selected HFs was 520. The list of the facilities with their allocated sample size were as follows: Ayder Comprehensive Specialized Hospital (143x283/520=78), Mekelle General Hospital (171x283/520=93), Quiha General Hospital (42X283/520=23), Aynalem Health Center (9x283/520=5), Semien Health Center (46x283/520=25), Adishumdhun Health Center (28x283/520=15), Adiha Health Center (11x283/520=6), and Kasech Health Center (70x283/520=38). Every woman who attended the PMTCT service during the data collection period was included in the study until the required sample size was achieved because the number of PMTCT attendees in these HFs were small in number. Hence, it was impossible to apply a further sampling technique.

## Variables of the study and their measurements

### Dependent variable

Compliance with the WHO-recommended infant feeding practice was the dependent variable, and it was measured dichotomously as compliance with the WHO-recommended infant feeding practice and non-compliance with the WHO-recommended infant feeding practice. When the mothers were answer "yes" to all the following questions: initiation of breast feeding within one hour, no pre-lacteal feeding, exclusive breastfeeding for the first 6 months of delivery, introduction of complementary food at 6 months, heat treated expressed breast milk, no wet nursing, and continued breast feeding at least for 12 months were considered as they have compliance with the WHO-recommended infant feeding practice [16]. Whereas, non-compliance with the WHO-recommended infant feeding practice is defined in this study as when mothers answer "No" to at least one of the above questions [16].

### Independent variables

Socio-demographic variables such as maternal age, marital status, occupation of mother and husband, maternal and husband educational status; medical status of the women and infant such as maternal CD4 count, mothers WHO clinical

 

stage, breast condition, presence of mouth ulcer in infants; maternal health factors such as having ANC follow up, place of delivery, postnatal follow up, and mode of delivery; others include knowledge and attitude of the mothers on infant feeding practice.

Therefore, some of the predictor variables, such as attitude, knowledge, WHO clinical stage, exclusive breastfeeding, mixed feeding, and exclusive replacement feeding, are operationally defined as follows. **Attitude:** there was a 3-point Likert scale with 5 attitudinal questions. A participant was given a score of 2 if they responded as agree, 1 for no opinion (neutral), and 0 for those who responded as disagree. Favorable attitude in this study was defined as when participants scored a mean or above, whereas participants who scored below the mean was considered to have an unfavorable attitude [42]. **Knowledge:** knowledge was assessed based on the following 10 dichotomized variables in the form of Yes/No answer questions. The statement questions include, having knowledge on the possibility of vertical transmission of HIV from mother to fetus/infant during pregnancy, delivery and breastfeeding, knowledge on the importance of breastfeeding for their child, avoiding of pre-lacteal feeding, breastfeeding initiation within one hour, the start of complementary feeding at six months, breastfeeding at least for 12 months, heat treatment for expressed breast milk and knowledge on that ART reduces risk of HIV transmission. The question items were gathered from different literature and contextualized to our study [24,26,42,43,44]. Therefore, knowledge of mothers was measured as "good knowledge" when participants were scored at or above the mean for the knowledge questions, whereas participants scored below the mean were considered to have "poor knowledge" [42]. Data for the WHO clinical staging for the mothers were taken from their charts and for some charts that missed staging, the team had staged using the WHO clinical staging system [45]. **Exclusive breastfeeding:** Giving the infant no other food or drink, apart from breast milk (including expressed breast milk), except for drops or syrups consisting of vitamins, mineral supplements, or prescribed medicines for up to six months. **Mixed breastfeeding:** Breastfeeding with the addition of fluids, solid feeds, and non-human milk in the first 6 months of age. **Exclusive replacement feeding:** The process of feeding a child who is not receiving breast milk with a diet that provides all the nutrients the child needs, until the child is fully fed on family foods. Recommended infant feeding practice: those who practiced either exclusive breastfeeding or exclusive replacement feeding up to six months of age [16].

## Data collection tool and procedures

The questionnaire was first prepared in English, which was adapted from similar topics published literatures (17–19) and contextualized to the specific objective of this study. Then translated to the local language (Tigrigna) and back to English by language experts to check its consistency. The tool had the following sections: Scio-demographic variables, obstetrical and maternal health variables, medical-related variables, knowledge and attitude variables. Two degree and 8 diploma holder midwives who were not staffed in these facilities were recruited as supervisors and data collectors, respectively. Training was given over two days by the principal investigator on how to conduct the data collection process and in making clarity about the variables. Some modifications were made after analyzing the pre-test results before the actual data collection began. After pretest data were collected using a semi-structured questionnaire through a face-to-face interview technique, once they got the service in a private room. The data were collected during work time from Monday to Friday.

## Data quality control

Data quality was ensured through translation of the English version tool into the local language (Tigrigna) for better understanding by the study participants and data collectors. The data collectors and supervisors could read and speak the local language fluently, and they were midwives by profession with experience in data collection. Training was given for two days by the principal investigator before starting the actual data collection. The tools were pre-tested among 5% of the sample size in Wukro General Hospital. Collected data were checked daily for completeness and consistency by supervisors and weekly by the principal investigator for cross-check.

## Data analysis and management

The returned hard copy questionnaires were checked visually for completeness, cleaned manually, and then coded and entered into SPSS version 21 software for analysis. Descriptive analysis was performed to summarize the data in the form of percentages, frequency, mean, and standard deviation. Logistic regression analyses were performed to assess the associations between variables. Hence, bi-variable analysis was performed to identify the relationship between dependent and explanatory variables. Variables that have made an association at a p-value of less than 0.25 were included in the final multivariable analysis. The model of fitness was checked by Hosmer-Lemeshow goodness of fit with a p-value of >0.05 and multicollinearity was analyzed using the variance inflation factor to look for interaction among the independent variables. The odds ratio was used to determine the direction and strength of the association. The Significance level and association of variables were tested by using a 95% confidence interval (CI) at a p-value less than 0.05 as cut off point. Tables, graphs, and text were used to organize and present the data.

## Ethical statement

Ethical clearance was obtained from the institutional ethical review board of Mekelle University, College of Health Sciences, with a reference number of MU-IRB1810/2020. Then, a letter of permission was obtained from all relevant authorities of the Tigray Regional Health Bureau and the study Health facilities. After the purpose and objective of the study had been explained, written informed consent was obtained from each study participant prior to enrollment in the study. Participants were informed that participation in this study was voluntary and were further informed about the benefits and risks of participation in this study. The names of respondents were excluded from the questionnaire, and all data were stored in secured place starting from the data collection site up to the analysis stage.

## Results

### Socio-demographic characteristics

A total of 283 HIV-positive mothers who had HIV-exposed infants participated in this study, with a response rate of 100%. The mean age of the respondents was 27.1 years (SD 4.80), which ranged from 18 to 42 years old. Out of the total study participants, about 94% (266) were Tigrian, followed by Amhara 3.53% (10), and Afar 2.5% (7) sequentially. About 76% (214) of participants were orthodox followers, and the rest were accounted for by Muslim 16.7% (47), Catholic 4.9% (14), and Protestant 2.8% (8). The majority of the respondent 82% (233), were married, and the rest were; 8% (24) divorced, 5% (13) single, and 5% (13) widowed. About 73% (207) of participants were from urban areas. Nearly one-third of the respondents, 30% (85), attended elementary school, 20% (56) completed high school, 15% (44) attended college and above, 28% (78) were illiterate, and 7% (20) had no formal education but were able to write and read. More than half, 51% (146), of the mothers' occupation was housewife, followed by daily laborer, 25% (70), private business, 17% (48), and government employee, 7% (19). With regard to the infants, the mean age of the infants was 9.1(SD 4.1) months, and about 42% (120) of the infants were found between the ages of 6–12 months, 31% (87) between the ages of 13–18 months, and 27% (76) less than 6 months. More than half, 51% (145), of the infants were females (**further refer to** Table 1).

### Obstetric history

Approximately 96% (271) of the mothers had at least one antenatal follow-up. Among these, around 8% (23) had 1–2 frequency of contacts, 16% (44) had 3–4 contacts, and 76% (204) had more than 4 contacts. Almost 99% (279) of the mothers were delivered at health facilities, of which 64% (179) were delivered at a hospital. With regard to mode of delivery, 80.6% (228) were delivered vaginally, and the rest 19.4% (55) were delivered by caesarean section (CS). Of which the vaginal delivery, around3.5% (8) were assisted with episiotomy. Post-natal coverage was 76% (214) in this study, of

 

**Table 1. Socio demographic characteristics of HIV-positive mothers enrolled in PMTCT service in governmental health facilities in Mekelle town, 2021. (n = 283).**

| Variables | Frequency (n) | Percent (%) |
|---|---|---|
| **Age of mothers (years)** | | |
| 18-24 | 104 | 36.7 |
| 25-29 | 97 | 34.3 |
| 30-34 | 60 | 21.2 |
| ≥ 35 | 22 | 7.8 |
| **Residence** | | |
| Urban | 207 | 73.1 |
| Rural | 76 | 26.9 |
| **Husband's educational status(n = 233)** | | |
| Unable to write and read/ illiterate | 35 | 15 |
| Able to write and read | 12 | 5.2 |
| Elementary | 62 | 26.6 |
| High school | 72 | 30.9 |
| College and above | 52 | 22.3 |
| **Husband's occupational status(n = 233)** | | |
| Daily labor | 71 | 30.5 |
| Farmer | 37 | 15.9 |
| Government employed | 38 | 16.3 |
| Merchant | 87 | 37.3 |

which about 78% (168) had 1–2 frequency of post-natal care visits and 22% (46) had 3 or more visits. Nearly 95% (268) of the mothers had been counseled about infant feeding options by health professionals.

## Medical condition of the mothers and infants

More than three out of five, 64% (180) of mothers had a CD4 count of >500 cells/mm3, 29% (83) had 200–500 cells/mm$^3$, and 7% (20) had less than 200 cells/mm$^3$. As Fig 1 depicts, more than four out of five mothers, 81% (228), were found on WHO clinical stage I. Almost 98% (277) of the infants took antiretroviral (ARV) prophylaxis immediately after delivery. Among the total infants, 9% (24) had developed mouth ulcers. A proportion of 8% (22) of the mothers encountered breast problems such as breast engorgement, cracked nipple, burning and tingling, and sore nipples in the first six months after delivery. The result showed that most of the respondents, 85% (240), disclosed their HIV status, of whom 74% (177) disclosed it to their partners, 16% (38) to close family members, and 10% (25) to their friends. An antibody test for HIV was done for about 91 infants, of which 98% (89) had negative results, and only 2% (2) were positive.

## Infant feeding practice among women enrolled in PMTCT

From the total of 283 HIV-positive mothers, about 79% (224) of them had compliance with the WHO-recommended infant feeding practices, whereas the rest, 21% (59), exercised inappropriate feeding practices. The majority, 91% (257) of mothers had ever breastfed their babies, and almost all of them, 99% (245) had given breast milk to their infants within 1 hour. More than four out of five, 86% (242) of mothers had exclusive breastfeeding, whereas 9% (26) and 5% (15) of mothers had exclusive replacement and mixed feeding practice, respectively. None of the respondents was given pre-lacteal feeding to their newborns. Among mothers who practiced replacement feeding, the major reasons for doing so were 65% (17) fear of HIV transmission through breast milk, and 35% (9) due to maternal illness. The reasons for mixed feeding

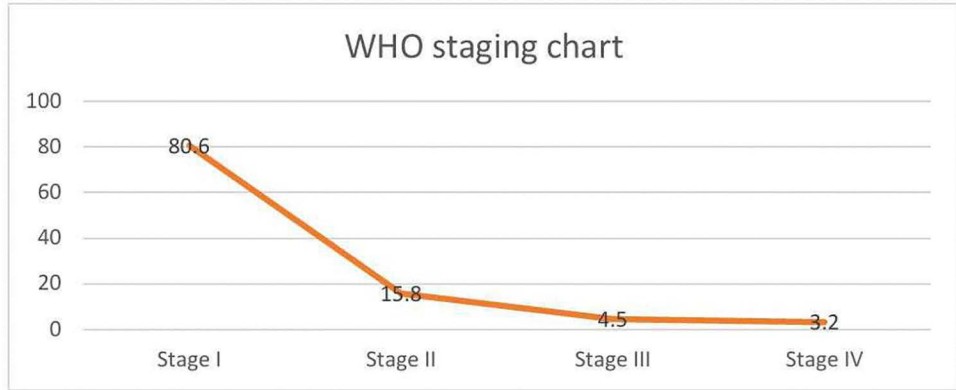

**Fig 1. The percentage distribution of mothers across WHO HIV/AIDS staging among women enrolled in PMTCT in Mekelle, Ethiopia, 2021.**

were that about 47% (7) considered that only breast milk feeding was insufficient for their babies, 33% (5) due to the husband 's opinion, and 20% (3) due to neighbors' advice. This study illustrated that neither expressed breast milk nor wet nursing was practiced. About 72% (204) of women introduced complementary feeding right at six months, 2% (5) started before six months and the rest started after 6 months. During the study period, about 12% (33) of mothers discontinued breastfeeding, of which 21% (7) were discontinued before they finished the infancy period (before 1 year). The reasons were 71% (5) due to fear of mother-to-child-transmission and 29% (2) considering no need for breastfeeding beyond that duration.

### Knowledge of HIV positive mothers towards MTCT and infant feeding practice

A high proportion of mothers, 91.5% (259) in this study, had sufficient comprehensive knowledge about appropriate infant feeding practice. The majority of the mothers, 90% (236), knew that HIV can be transmitted from mother to child, and about 90% (236) of them also knew that HIV can be transmitted during breastfeeding. More than three-fifths of the mothers, 71% (202), knew that the recommended time for exclusive breastfeeding was 6 months (**further refer to** Table 2).

### Attitude of HIV positive mothers toward infant feeding

The study revealed that the majority, 71% (201) of the mothers had a favorable attitude towards the recommended feeding option for the first six months. About 86% (244) of HIV-positive mothers had a favorable attitude towards infant feeding in general (**refer to** Table 3).

### Factors associated with infant feeding practice

The results of bi-variable logistic regression analysis showed that the following variables: residence, ANC follow-up, mode of delivery, postnatal follow-up, exposure to counseling about infant feeding options, mothers' breast diseases, HIV disclosure status, and attitude had a significant association with the WHO-recommended infant feeding practice for HIV exposed infants at p value of <0.25. Hence, after controlling for possible confounders in multivariable logistic regression analysis, ANC follow-up, mode of delivery, HIV disclosure status, and attitude were significantly associated (at p- value <0.05) with the WHO-recommended infant feeding practice. Those mothers who had ANC follow-up were 5 times more likely to follow the appropriate infant feeding practice than those who had no ANC follow-up [AOR = 4.92 [95% CI: 1.17-20.66]]. Mothers who gave birth through spontaneous vaginal delivery (SVD) were 3 times more likely to practice appropriate feeding practices than those who delivered by CS [AOR = 3.04 [1.46-6.32]]. Mothers who disclosed their HIV status

**Table 2. Knowledge of HIV-positive mothers towards infant feeding enrolled in PMTCT service in governmental health facilities in Mekelle city, 2021.**

| Variable | Category | Frequency (283) | Percentage (%) |
|---|---|---|---|
| Know HIV can be transmitted during pregnancy | Yes | 217 | 77 |
| Know HIV can be transmitted during delivery | Yes | 225 | 79.5 |
| Know HIV can be transmitted during breastfeeding | Yes | 236 | 90 |
| Knew breastfeeding is important to their infant | Yes | 259 | 91.5 |
| Knew to avoid pre-lacteal feeding | Yes | 240 | 85 |
| Knew breastfeeding initiation within 1 hour | Yes | 178 | 63 |
| Knew complementary feeding to be started at six months | Yes | 155 | 55 |
| Knew the importance of breastfeeding for 1 year and beyond | Yes | 172 | 61 |
| Knew expressed breast milk should be heat-treated | Yes | 180 | 64 |
| Know drug treatment reduces MTCT of HIV | Yes | 145 | 51 |

**Table 3. Attitude of HIV-positive mothers towards breastfeeding enrolled in PMTCT service in governmental health facilities in Mekelle city, 2021.**

| Variable | Category | Frequency | Percent |
|---|---|---|---|
| Replacement feeding should be given consistently for six months. | Agree | 234 | 82.7 |
|  | No opinion | 17 | 6 |
|  | Disagree | 32 | 11.3 |
| Exclusive breastfeeding for the first six months is the best choice for infant feeding. | Agree | 201 | 71 |
|  | No opinion | 58 | 20.5 |
|  | Dis agree | 24 | 8.5 |
| HIV/AIDS can transmit from mother-to-child through breastfeeding. | Agree | 214 | 75.6 |
|  | No opinion | 52 | 18.4 |
|  | Disagree | 17 | 6 |
| Mixed feeding can increase the risk of HIV/AIDS transmission. | Agree | 216 | 76.3 |
|  | No opinion | 44 | 15.5 |
|  | Dis agree | 23 | 8.2 |
| When the CD4 count is high, the risk of HIV transmission through breastfeeding is decreased. | Agree | 214 | 75.7 |
|  | No opinion | 44 | 15.5 |
|  | Dis agree | 25 | 8.8 |

to their spouse were 3 times more likely to have appropriate feeding practice than those who didn't disclose their status [AOR = 2.72[95% CI:1.23-6.02]]. Mothers with a favorable attitude towards breastfeeding were 4 times more likely to have appropriate feeding practice than their counterparts [AOR = 4.31 [95% CI:1.89-9.80]] (further **refer** Table 4 **below**).

## Discussion

This study mainly focused on assessing the magnitude of appropriate infant feeding based on the WHO recommendation and what are the factors influencing infant feeding practice among HIV-positive mothers enrolled in PMTCT service in health facilities. The study included children less than 18 months old who are HIV-exposed infants and used a comprehensive tool to assess infant feeding practices. Therefore, this study came up with notable findings, such as the proportion of infant feeding modalities, including the proportion of exclusive breastfeeding, mixed feeding, replacement feeding, and overall, the proportion of appropriate infant feeding practice according to the 2016 WHO guideline recommendation

**Table 4. Multivariate logistic regression analysis upon appropriate feeding practice of HIV-positive mothers enrolled in PMTCT service in governmental health institutions in Mekelle city, 2021.**

| Variables | Appropriate Infant Feeding practice | | COR (95% CI) | AOR (95% CI) |
|---|---|---|---|---|
| | Yes n (%) | No n (%) | | |
| **Residence** | | | | |
| Urban | 171(82.6%) | 36(17.4%) | 2.06 [1.123-3.783] | 1.82 [.91-3.62] |
| Rural | 53(69.7%) | 23(30.3%) | 1 | 1 |
| **ANC follow up** | | | | |
| Yes | 219(80.8%) | 52(19.2%) | 5.90 [1.80-19.32] | 4.92 [1.17-20.66] * |
| No | 5(41.7%) | 7(58.3%) | 1 | 1 |
| **Mode of delivery** | | | | |
| C-section | 35(63.6%) | 20(36.4%) | 1 | 1 |
| SVD | 189(83.0%) | 39(17.0%) | 2.83[1.47-5.43] | 3.04 [1.46-6.32] * |
| **PNC follow up** | | | | |
| Yes | 176(82.2%) | 38(17.8%) | 2.03 [.92 -7.93] | 1.92 [.94-3.93] |
| No | 48(69.6%) | 21(30.4%) | 1 | 1 |
| **Counseled on feeding options** | | | | |
| Yes | 215(80.2%) | 53(19.8%) | 2.70 [.92-7.93] | 2.15 [.55-8.39] |
| No | 9(60%) | 6(40%) | 1 | 1 |
| **Breast disease** | | | | |
| Yes | 13(59.1%) | 9(40.9%) | 1 | 1 |
| No | 211(80.8%) | 50(19.2%) | 2.92 [1.18-7.22] | 2.64 [.87-8.03] |
| **Disclosure status** | | | | |
| Yes | 197(82.1%) | 43(17.9%) | 2.72 [1.35-5.47] | 2.72 [1.23-6.02] * |
| No | 27(62.8%) | 16(37.2%) | 1 | 1 |
| **Attitude** | | | | |
| Favorable | 204(83.6%) | 40(16.4%) | 4.85 [2.37-9.89] | 4.31 [1.89-9.80] * |
| Unfavorable | 20(51.3%) | 19(48.7%) | 1 | 1 |

**Note:** COR = crude odds ratio, AOR = adjusted odds ratio, ANC antenatal care, SVD = spontaneous vaginal delivery and C-section = caesarean section, * p value<0.005

for HIV-exposed infants. In addition, this study also identified several variables that were associated with the outcome variable, such as having ANC follow-up, mode of delivery, HIV disclosure status, and attitude towards appropriate infant feeding practice at a p-value of <0.05.

In this study, the proportion of recommended infant feeding practice among women living with HIV was 79%, which was consistent with several findings of studies conducted in various parts of Ethiopia, including Addis Ababa [26], Adama [24], Debere-Markos [28], Bahir-Dar [46], and a systematic review and meta-analysis from Ethiopia [33], where 82.6%, 85.5%, 85.8%, 83.5%, and 82.8% of mothers had the recommended safe infant feeding practice, respectively. However, the proportion of safe infant feeding practice in this study was slightly lower than that of two studies conducted in Gondar and Shashamane, where 89.5%, 94%, and 89.1% of mothers living with HIV had appropriate infant feeding practice, respectively [25,29,32]. The possible justification for the difference between the results could be due to the time elapsed between the studies, sample size, and socio-demographic differences. Over the course of time, access to information from health care providers and technology become advanced, which in turn increases knowledge of mothers about HIV transmission and care of HIV-exposed infants. However, socio-demographic variables such as educational status, residency, and income status can also affect the knowledge and practice of infant feeding, which leads to this difference. Moreover,

health care system factors and health-seeking behaviors might be varying from region to region across the country and continent. This can also be the reason that leads to a difference in the magnitude of practice. The sub-analysis result of this study also found that exclusive breastfeeding was 82%, which is comparable with the study findings of Gondar (83.8%), Southern part of Ethiopia, (85.6%) [47], Tigray (88.8%) [19], Debre-Markos (77.1%) [28], and Bahir-Dar (75.2%) [44]. However, it was higher than the study findings from two systematic studies conducted in Ethiopia (63.43%, 63.99%) [48, 49], Afar region (63.8%) [30], Southern Ethiopia (49.3%) [22], Somalia region (52%) [50], Kenya (71.4%) [51], Nigeria (61%) [52] and Zambia (30.1%) [53]. The possible reason behind this difference could be due to socio-demographic, dietary culture, study area/setting difference, time elapsed between the studies, and health care system differences between the counties and the study areas. For instance, two studies were conducted in the Afar and Somalia regions of Ethiopia, which are pastoralist regions where access to health services could be compromised due to their lifestyle and limited geographic access to health services. The breastfeeding culture of other African countries might be different from that of Ethiopia. In addition, health policies and intervention strategies for HIV-exposed infants could also be different from the country Ethiopia. In Ethiopia, breastfeeding is a deeply rooted cultural practice among mothers.

The prevalence of HIV-positive mothers practiced exclusive replacement feeding (ERF) for the first six month was 9%, which is in line with findings reported from Bahir-Dar (13.9%) [44], Debre-Markos (8.5%) [28], Shashamane (9.3%) [32], Southern Ethiopia (6.1%) [47], Oromia region (6.3%) [24], and Kenya (10.4%) [51]. However, other studies conducted in Ethiopia revealed a lower proportion of exclusive replacement feeding than the magnitude reported in this study. For instance, reports from Addis Ababa [26], two studies from Tigray [19,23], and Gondar [31] showed that 3.8%, 3.4%, 4.6%, and 4.5% of mothers living with HIV had exclusive replacement feeding, respectively. This might be due to the fact that more than 90% of women in this study had sufficient knowledge of infant feeding practices and how HIV can be transmitted. In addition, approximately 85% of participants of this study received counseling about the recommended infant feeding practice from health care providers. Hence, these women might realize that the risk of breastfeeding as one way for vertical HIV transmission, which leads to using exclusive replacement feeding as an option. In contrast, this finding is lower in magnitude compared to findings of studies conducted in Bahir-Dar city (47.4%) [46], a systematic and meta-analysis study in Ethiopia (16.13%) [49], the Southern region of Ethiopia (19.9%) [22], Nigeria (26%) [52], and South Africa (56.8%) [54]. The reason could be that women in this study might not have the financial capacity to afford formula feeding or, in general, they might not fulfill the AFFAS (acceptable, feasible, affordable, sustainable, and safe) criteria to use exclusive replacement feeding as a feeding option. Therefore, women might consider exclusive breastfeeding as an option rather than exclusive replacement feeding. On the other hand, countries like Nigeria and South Africa might encourage their citizens to practice exclusive replacement feeding as a feeding option to prevent the vertical transmission of HIV infection from mother to child during breastfeeding. Because these countries are economically better off than Ethiopia, citizens living in these countries could have the capacity to afford formula feeding as compared to citizens living in Ethiopia. The inherent practice of breastfeeding as a culture in Ethiopia might make it difficult for a mother to choose exclusive replacement feeding as a feeding option, and sometimes this could be an official announcement of their HIV status to the community.

In this study, the proportion of mothers who practiced mixed feeding was 5%, which is comparable with studies done in Tigray (6.6%, 6.3%) [19,23], Oromia, 8.3% [24], and the Southern part of Ethiopia (8.3%) [47]. However, the proportion is lower than the study done in Bahir-Dar (10.9%) [44], Addis Ababa (17.4%) [26], Debre-Markos (14.2%) [28], Afar (36.2%) [30], Gondar (10.5%, 21.6%) [25, 31], the Sothern Ethiopia (30.8%) [22], systematic and meta-analysis studies from Ethiopia (23.11%, 20.95%) [48,49], and Kenya (18.2%) [51]. The possible justification behind this difference could be that the majority of the respondents received counseling about the recommended feeding options from health care providers in this study. In addition to this, participants of this study also demonstrated sufficient knowledge about feeding options and the risk of HIV transmission from mother to their newborns during breast feeding period.

This study revealed that neither expressed breast milk nor wet nursing was practiced. Similar finding was found in studies conducted in Mekelle and Gondar cities support this finding [23,25]. This could be that manually expressed and heat-treated breast milk is not considered a normal/acceptable practice culturally in Ethiopia. Moreover, nowadays wet-nursing might not be practiced in the community due to the risk of cross- transmission of HIV infection from mother to child. Mothers who had ANC follow-up were five times more likely to follow the recommended feeding practice than those who had no ANC follow-up. Similarly, a study finding from the Oromia region (Ethiopia) found that women living with HIV and having ANC contact were 0.05 times less likely to practice inappropriate infant feeding [24]. Similar Studies conducted in different parts of Ethiopia, for example, Debre-Markos, Gondar, Somalia region, Bahir Dar, and a systematic and meta-analysis study from Ethiopia [28,31, 33, 46, 50], also reported that women living with HIV and having ANC follow-up were more likely to practice appropriate infant feeding as compared to their counterparts. This might be due to the fact that mothers who had ANC follow-up could have a high chance of exposure to counseling on recommended feeding options and how to prevent vertical HIV transmission from mother to children, as compared to their counterparts. This suggestion, supported by studies conducted in Shashamane (Ethiopia), indicated that women who received counseling about infant feeding from health professionals were 4.5 times more likely to practice appropriate infant feeding [32]. Other studies conducted in Ethiopia (systematic and meta-analysis, Oromia, Somalia, and Tigray regions) and Nigeria also corroborated that exposure to counseling about infant feeding during ANC was significantly associated with the WHO-recommended infant feeding practice [19,23,24,48,52]. This study also showed that 95% of mothers received counseling about feeding options for HIV exposed infants during ANC.

This study showed that mothers who gave birth through the vagina were 3 times more likely to practice appropriate feeding practices than those who delivered by CS. This finding is consistent with findings reported from Addis Ababa [26], where mothers who gave birth through CS were more likely to practice replacement feeding than women who delivered through the vagina and 80% less likely to use exclusive breast feeding according to the WHO recommendation [55]. Another piece of evidence from four Southeast Asian countries supports the finding that babies born by CS were less likely to have exclusive breastfeeding [56]. Numerous studies reported that CS delivery increases the odds of late initiation of breastfeeding [34, 57, 58]. This is because women who had CS have postoperative pain and discomfort, and an anesthetic effect. These conditions may require women to stay longer in the recovery room before making their first contact with their newborn until recovered from these effects, which in turn delays initiation of breastfeeding after birth [59]. In addition, mothers may also encounter difficulties achieving a comfortable breastfeeding position due to the CS wound. On the other hand, infants born by CS, particularly due to emergency conditions, are more likely to have respiratory problems that necessitate admission to the intensive care unit immediately. This delay can play a significant role in the establishment of early breastfeeding practices, which can negatively lead to unsuccessful exclusive breastfeeding. Several studies revealed that as women delayed initiating breastfeeding within one hour of birth, the success of exclusive breastfeeding for the first six months, including maintaining breastfeeding for longer periods, decreased [60, 61, 62]. However, one study suggests that, in the presence of counseling and adequate support, CS delivery is not necessarily a barrier to timely breastfeeding initiation or establishing exclusive breastfeeding [20].

This study showed that mothers who disclosed their HIV status to their spouses were 3 times more likely to follow the recommended feeding options than those who did not disclose their HIV status. This finding is consistent with the studies done in Debre-Markos [28], Afar [30], Gondar [25, 29], systematic and meta-analysis study from Ethiopia [33, 48], Bahir-Dar [46], and Oromia [24]. The possible explanation behind this, disclosure of HIV status to a spouse or other family member, might encourage couples to discuss feeding options freely and offer psychological relief, to implement the recommended WHO infant feeding practice for HIV-exposed infants, as compared to women who do not disclose their status. Likewise, disclosure can also strengthen relationships among family members. This, in turn, promotes and enables the family to support and encourage the mother living with HIV to accept and implement medical advice and care. On the other hand, mothers choosing to feed their children through formula feeding may have a fear of stigmatization in

many cultures, which can affect adherence to replacement feeding [34]. Moreover, even in the absence of stigma, women may have financial problems with affording supplies for formula feeding, which can lead to mixed feeding [63].

This study found that mothers who have a positive attitude towards appropriate infant feeding practice were 4 times more likely to practice the recommended infant feeding practice. This is consistent with studies done in Tigray, Ethiopia, and Addis Ababa, the capital city of Ethiopia [19,23 26]. This might be because, as mothers have a favorable attitude, the likelihood of deciding to implement the recommended way of infant feeding practice becomes high.

## Strength and limitations

Despite its importance, this study is not without limitations. Some of the limitations are: there might be recall bias when respondents requested information, since the data were collected retrospectively, and due to the cross-sectional nature of this design, causality and effect couldn't be determined in this study. Furthermore, this study was conducted among mothers attending PMTCT services in health institutions. As a result, the findings may not be generalizable to mothers utilizing services from private and non-governmental institutions, as well as to the entire region. Despite these limitations, information found from this study is crucial and can serve as an input for the decision-making process about infant feeding options for HIV-exposed infants among women living with HIV in the region, as well as nationwide. It also plays a crucial role in providing information about this vulnerable group to ensure health service equity in the community, thereby achieving the Sustainable Development Goals.

## Conclusion and recommendation

This study found that a large proportion of HIV-positive mothers had followed the recommended infant feeding practice. However, a significant number of mothers living with HIV are still practicing mixed types of feeding for their HIV-exposed infants. This type of feeding practice is a driving factor in producing an HIV-positive new generation. ANC follow-up, mode of delivery, HIV disclosure status, and attitude towards feeding practice are the major identified predictors for the recommended infant feeding practice in this study. Therefore, frontline health care workers should strengthen and emphasize the importance of counseling and care services in PMTCT regarding the role of ANC attendance, disclosing HIV status to a spouse and family members, counselling and support about breastfeeding for women delivered by CS. Moreover, these services should also be provided to the general population during the provision of maternal health services throughout the continuum of care. The regional health bureau and other stakeholders should provide resources and supportive supervision to strengthen the counseling system and care provision in HIV diagnosis and treatment centers in general and at PMTCT service in particular.

## Supporting information

**S1 Data. SPSS data used to conclude the findings of this study.**
(SAV)

## Acknowledgments

We would like to express our deepest gratitude and appreciation to Mekelle University College of Health Sciences Department of Midwifery, for their coordination throughout the process of this thesis work and for their financial support. Also, our special thanks go to the participants of this study for their time.

## Author contributions

**Conceptualization:** Haftom Tadesse Alemayohu.

**Data curation:** Haftom Tadesse Alemayohu.

Formal analysis: Solomon Weldemariam Gebrehiwot, Haftom Tadesse Alemayohu, Hadgay Hagos.

Funding acquisition: Haftom Tadesse Alemayohu.

Investigation: Haftom Tadesse Alemayohu.

Methodology: Haftom Tadesse Alemayohu.

Resources: Haftom Tadesse Alemayohu.

Software: Haftom Tadesse Alemayohu.

Supervision: Solomon Weldemariam Gebrehiwot, Hadgay Hagos.

Validation: Solomon Weldemariam Gebrehiwot, Hadgay Hagos.

Visualization: Solomon Weldemariam Gebrehiwot, Hadgay Hagos.

Writing – original draft: Solomon Weldemariam Gebrehiwot.

Writing – review & editing: Haftom Tadesse Alemayohu, Hadgay Hagos.

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
