## [Editor Report · Decision Letter 0]

11 Sep 2025

PGPH-D-25-01884

Compliance with the WHO-recommended infant feeding practices among HIV positive mothers at Public Health Facilities of Mekelle, Tigray, norther Ethiopia

Dear Dr. Gebrehiwot,

Thank you for submitting your manuscript to PLOS Global Public Health. I have not sent this out for peer review yet, but as an editor I have some comments.

We look forward to receiving your revised manuscript.

Kind regards,

Abram L. Wagner, PhD, MPH

Academic Editor

You have a sample size of 283. That might be fine for your study, but I would simplify how you describe your results. I'd recommend rounding to the one's place (e.g., mention 73% instead of 72.9%), and having two decimal places for ORs (e.g., 2.56 instead of 2.558). That just makes things cleaner.

In the methods section, I recommend adjusting the order of presentation. At present, you first list independent and dependent variables and only afterward describe how they are measured. It would be clearer if, after introducing the dependent variables, you immediately describe their measurement, and then do the same for the independent variables.

Regarding the knowledge score, you note a cutoff of 10 based on the scoring of individual items. Please clarify whether this threshold was chosen ad hoc or whether it is supported by prior literature.

On the use of abbreviations, I suggest a more cautious approach. Some acronyms, such as HIV, are of course universally recognized, but others (e.g., HIV-EI, ERF, AFFAS) are not as familiar to a general readership. Writing them out or minimizing their use would improve readability and reduce the need for readers to flip back and forth in the text.

For the WHO staging figure, it would help to provide more methodological detail in the methods section explaining how staging was assessed.

In the results section, I recommend reversing the order in which you present counts and percentages. Presenting percentages first in the text, followed by raw numbers in parentheses, will make the findings easier to follow.

In Table 2, for dichotomous variables you only need to report one category (e.g., “Yes”), since the complementary “No” values are redundant and add unnecessary length.

In your description of the multivariable model, please avoid calling the preliminary inclusion threshold of p = 0.25 “significant.” This threshold is appropriate for model development, but it should not be presented as a measure of statistical significance.

In Table 4, there is a small typo (“C-O-A”) that should be corrected to “C-O-R.”

Finally, with regard to your figures, I encourage you to reconsider whether they add value. The pie chart does not present information in a particularly clear way, and the WHO staging chart may duplicate results that could be more concisely conveyed in a table or brief paragraph. Simplifying these visual elements would strengthen the overall presentation.

Journal Requirements:

1. Please amend your online detailed Financial Disclosure statement. This is published with the article. It must therefore be completed in full sentences and contain the exact wording you wish to be published.

a) State the initials, alongside each funding source, of each author to receive each grant, if applicable. For example: "This work was supported by the National Institutes of Health (####### to AM; ###### to CJ) and the National Science Foundation (###### to AM)."

For more information, please go to our submission guidelines:

https://journals.plos.org/globalpublichealth/s/submission-guidelines#loc-financial-disclosure-statement

2. Please ensure that the funders and grant numbers match between the Financial Disclosure field and the Funding Information tab in your submission form. Note that the funders must be provided in the same order in both places as well.

3. Please update your online Competing Interests statement. If you have no competing interests to declare, please state: “The authors have declared that no competing interests exist.”

4. Please provide separate figure files in .tif or .eps format only and ensure that all files are under our size limit of 10MB.

5. Please ensure that you refer to Figure 1 in your text as, if accepted, production will need this reference to link the reader to the figure.
---

## [Decision Letter · Decision Letter 1]

15 Dec 2025

PGPH-D-25-01884R1

Compliance with the WHO-recommended infant feeding practices among HIV positive mothers at Public Health Facilities of Mekelle, Tigray, norther Ethiopia

Dear Dr. Gebrehiwot,

Thank you for submitting your manuscript to PLOS Global Public Health. After careful consideration, we feel that it has merit but does not fully meet PLOS Global Public Health’s publication criteria as it currently stands. Therefore, we invite you to submit a revised version of the manuscript that addresses the points raised during the review process.

Please respond to both reviewers - and if you feel like there is overlap in their comments, feel free to reference your response across reviewer comments.

We look forward to receiving your revised manuscript.

Kind regards,

Abram L. Wagner, PhD, MPH

Academic Editor

Journal Requirements:

Additional Editor Comments (if provided):

Reviewers' comments:

Reviewer's Responses to Questions

**Comments to the Author**

Reviewer #1: (No Response)

Reviewer #2: (No Response)

publication criteria?

Reviewer #1: Yes

Reviewer #2: Yes

3. Has the statistical analysis been performed appropriately and rigorously?

Reviewer #1: Yes

Reviewer #2: Yes

4. Have the authors made all data underlying the findings in their manuscript fully available (please refer to the Data Availability Statement at the start of the manuscript PDF file)?

Reviewer #1: Yes

Reviewer #2: Yes

5. Is the manuscript presented in an intelligible fashion and written in standard English?

Reviewer #1: No

Reviewer #2: Yes

Reviewer #1: Thank you very much for giving me the opportunity to review this paper. Despite all the strides made in the control of HIV, it continues to be a significant burden on healthcare systems, communities and individuals. I found this paper to be an important contribution to the existing body of work on PMTCT. Please note that my comments are based on the revised version of the paper that is in tracked changes.

General

I found the paper to be well structured. However, I believe that the quality and readability of the paper will be greatly strengthened by having a thorough review of the language. For example, the lines 102-104 may read better if presented as “Hence, to create a HIV-free generation, countries need updated and consistent information as well as updated strategies on infant feeding practices among HIV- exposed infants.” Throughout the paper, there are many grammatical errors and issues with syntax.

Background

- Line 88-91. This sentence seems to be incomplete. The factors have been enumerated but what are these factors associated with precisely?

Sample size determination and sampling process

- Line 137-138: There is no need to put “41.3% vs 21.5%, respectively” in brackets.

- Line 138-142: Here, you mention seven factors associated with infant feeding practice. It will be helpful if these seven factors are listed because prior to this point in the paper, you did not specifically list these seven factors that were of particular interest to you.

Results

- Line 288: Please consider adding the unit to the mean infant age. Should this be 9 months?

- Table 4: Episiotomy is not a form of delivery. Seeing that this was the only place it was mentioned in the paper, the authors should consider removing it from the table.

Discussion

- The use of the word “discrepancy” when comparing the results of this study with other studies (e.g. lines 435 and 449) may not necessarily be correct. I believe that the results of your study may be different from other studies but shouldn’t be taken as a discrepancy. The authors may wish to consider changing the use of the term discrepancy to perhaps difference.

Reviewer #2: The revised version of the manuscript is nearly perfect. There are few areas which I would like to request the authors to work on as follows:

ABSTRACT

Line 28 - I suggest to do the following:

o Edit “Compliance” to start with a small “c”; and

o Write WHO in its long form and the abbreviation in brackets as follows; “World Health Organization (WHO)”.

Line 36 – I suggest to replace “world health organization” with its abbreviation “WHO”.

BACKGROUND

Line 49 – I suggest to write HIV in its long form and the abbreviation in brackets as follows; “Human Immunodeficiency Virus (HIV)”.

Line 51 – I suggest to replace ““Human Immunodeficiency Virus (HIV)” with its abbreviation “HIV”.

Line 62 – I suggest to edit “UNAID” to be “UNAIDS”.

Line 75 – I suggest to write ART in its long form and the abbreviation in brackets as follows; “antiretrovirals therapy (ART)”.

Line 80 – I suggest to edit “also associates” to be “is also associated”.

Lines 84-85 – I suggest to insert “other” between “from & regions”.

Line 89 – I suggest to write ANC in its long form and the abbreviation in brackets as follows; “antenatal care (ANC)”.

Line 102 – I suggest to edit “update” to be “updated”.

Line 105 – I suggest to: (a) edit “update” (between Moreover & information) to be “updated; and (b) insert “is” between “Tigray & also”.

METHODS

Study setting, design and period

Line 118 – I suggest to insert “/” between HIV and AIDS.

DISCUSSION

Line 381 – I suggest to edit “vary” (between “be & from”) to be varying”.

CONCLUSION AND RECOMMENDATION

Line 507 – I suggest to insert “are” between HIV & still.

Line 514 – I suggest to replace “caesarean section” with its abbreviation “CS”.

**Do you want your identity to be public for this peer review?** For information about this choice, including consent withdrawal, please see our Privacy Policy

Reviewer #1: **Yes:**  Grace Mambula

Reviewer #2: **Yes:**  Eliudi Saria Eliakimu

---

## [Decision Letter · Decision Letter 2]

26 Jan 2026

Compliance with the WHO-recommended infant feeding practices among HIV-positive mothers at Public Health Facilities of Mekelle, Tigray, northern Ethiopia

PGPH-D-25-01884R2

Dear Mr. Gebrehiwot,

We are pleased to inform you that your manuscript 'Compliance with the WHO-recommended infant feeding practices among HIV-positive mothers at Public Health Facilities of Mekelle, Tigray, northern Ethiopia' has been provisionally accepted for publication in PLOS Global Public Health.

Best regards,

Abram L. Wagner, PhD, MPH

Academic Editor

Reviewer Comments (if any, and for reference):

Reviewer's Responses to Questions

**Comments to the Author**

Reviewer #1: All comments have been addressed

Reviewer #2: All comments have been addressed

publication criteria?

Reviewer #1: Yes

Reviewer #2: Yes

3. Has the statistical analysis been performed appropriately and rigorously?

Reviewer #1: Yes

Reviewer #2: Yes

4. Have the authors made all data underlying the findings in their manuscript fully available (please refer to the Data Availability Statement at the start of the manuscript PDF file)?

Reviewer #1: Yes

Reviewer #2: Yes

5. Is the manuscript presented in an intelligible fashion and written in standard English?

Reviewer #1: Yes

Reviewer #2: Yes

Reviewer #1: (No Response)

Reviewer #2: (No Response)

**Do you want your identity to be public for this peer review?** For information about this choice, including consent withdrawal, please see our Privacy Policy

Reviewer #1: **Yes:**  Grace Mambula

Reviewer #2: **Yes:**  Eliudi Saria Eliakimu
